# Classroom Movement Breaks Reduce Sedentary Behavior and Increase Concentration, Alertness and Enjoyment during University Classes: A Mixed-Methods Feasibility Study

**DOI:** 10.3390/ijerph18115589

**Published:** 2021-05-24

**Authors:** Casey L. Peiris, Gráinne O’Donoghue, Lewis Rippon, Dominic Meyers, Andrew Hahne, Marcos De Noronha, Julia Lynch, Lisa C. Hanson

**Affiliations:** 1School of Allied Health, Human Services and Sport, Physiotherapy, La Trobe University, Melbourne, VIC 3086, Australia; lewis.rippon@act.gov.au (L.R.); dominic.meyers@kieser.com.au (D.M.); A.Hahne@latrobe.edu.au (A.H.); Julia.Lynch@latrobe.edu.au (J.L.); 2UCD School of Public Health, Physiotherapy and Sports Science, University College Dublin, Belfield, Dublin 4, Ireland; grainne.odonoghue@ucd.ie; 3La Trobe Rural Health School, Holsworth Research Initiative, La Trobe University, Bendigo, VIC 3550, Australia; M.DeNoronha@latrobe.edu.au; 4Violet Vines Marshman Centre for Rural Health Research, La Trobe Rural Health School, La Trobe University, Bendigo, VIC 3550, Australia; L.Hanson@latrobe.edu.au

**Keywords:** sedentary behaviour, exercise, mental fatigue, students, universities

## Abstract

This mixed-methods study aimed to determine the feasibility of incorporating movement breaks into university classes in terms of acceptability (disruption, engagement, satisfaction), practicality (ease of scheduling and conducting breaks) and efficacy (sedentary time, concentration, alertness, enjoyment). Movement breaks of five to 10 min duration were scheduled after 20 min of sedentary time during 2-h classes. Classes without movement breaks were used as a comparison. Data were collected using surveys, objective physical activity monitoring and focus group interviews of students (*n* = 85) and tutors (*n* = 6). Descriptive statistics (quantitative data) and independent coding and thematic analysis (qualitative data) were completed. Students (mean age 23 ± 2 years, 69% female) actively engaged in movement breaks with no adverse events. Movement breaks were perceived to be beneficial for concentration, engagement and productivity. Timing of the break was perceived to be important to enhance the benefit and reduce disruption. Students preferred outdoor or competitive movement breaks. Students spent 13 min less time sitting (95%CI 10 to 17), took 834 more steps (95%CI 675 to 994) and had higher levels of concentration, alertness and enjoyment (*p* < 0.001) in classes with movement breaks compared to classes without. Classroom movement breaks are feasible and may be considered for incorporation into university classes to reduce sedentary behaviour and increase physical activity, alertness, concentration and enjoyment.

## 1. Introduction

High volumes of sedentary behaviour are associated with increased risk of poor health outcomes including all-cause mortality, obesity, cardiovascular disease, type 2 diabetes mellitus, cancer [1], depression [2] and anxiety [3]. An increasing proportion of young adults are presenting with signs and symptoms of excessive time spent sedentary including obesity, hypertension, dyslipidemia and metabolic syndrome [4,5]. Any waking behaviour with low energy expenditure (≤1.5 METs), including sitting at a desk, is classified as sedentary behaviour [6]. As university students may need to spend large amounts of their weekdays sitting in class and studying, they appear to be particularly sedentary compared to other young adults [7]. Based on objective assessment (accelerometry), university students spend an average of 9.8 to 10.7 waking hours each day in sedentary activities [7,8]. The sedentary time accrued while studying may be detrimental to the current and future health of university students. In addition to increasing physical activity, national guidelines now recommend regular breaks to interrupt sedentary time [9,10,11]. One such innovative opportunity may be in the university setting and specifically during seated classes. 

Breaking sedentary time with regular, brief, physical activity has general health benefits such as helping to control obesity and glycaemia in adults [12] and may improve cognitive capacity and academic performance in university students [8]. There is limited research on classroom movement breaks in the university setting, but initial research indicates university students had a positive attitude to movement breaks of three to five minutes duration conducted every 20 min during classes [13]. In another study, results indicated exercise breaks increased vigor and decreased fatigue levels for 20 min compared to no breaks [14]. A four-minute movement break conducted once in a 2-h lecture was perceived to be feasible and acceptable to students [15]. However, there is still concern that movement breaks may disrupt concentration and harm productivity [16]. 

Preliminary evidence suggests that movement breaks may be beneficial for university students, however, the feasibility of including movement breaks in university classes is unknown. Therefore, the primary aim of this study was to determine the feasibility of incorporating movement breaks into university classes. For this study, elements of feasibility considered were acceptability (levels of disruption, engagement and satisfaction), practicality (safety, ease of scheduling and conducting breaks) and efficacy (differences in sedentary time, concentration, alertness and enjoyment between classes with and without movement breaks). 

## 2. Materials and Methods

### 2.1. Design

A mixed-methods design using quantitative and qualitative approaches was conducted comprising of surveys, objective physical activity monitoring and focus group interviews. This design was chosen for triangulation [17] and mutual enhancement as the survey provided raw data on alertness, concentration and enjoyment, accelerometer data provided information on fidelity and efficacy, and focus group data provided participants’ experiences of movement breaks from the perspective of university students and tutors. Focus groups can generate rich, detailed data in the participants’ own words to explore their experiences of the movement breaks and were chosen over individual interviews as they assist participants to explore and clarify their views relative to other participants. The less formal nature of a focus group also allows participants to express their views candidly and spontaneously, while participant interactions within a focus group can also generate new ideas [18]. The Consolidated Criteria for Reporting Qualitative Research (COREQ) checklist [19] and the Strengthening Reporting of Observational Studies in Epidemiology (STROBE) statement [20] were used to ensure the rigor of reporting. University Human Research Ethics Committee approval (HEC19527) was obtained prior to commencement of the study. Participation was voluntary and all participants provided written informed consent. 

### 2.2. Participants

A convenience sample of final-year physiotherapy students from two campuses of one Australian university was invited to participate. Potential participants were informed about the study via email from an academic staff member not directly involved with the students. To be eligible, students needed to be concurrently enrolled in two subjects (‘chronic disease management’ (CLASS A) and ‘evidence-based practice’ (CLASS B)) in 2020 and be able to participate in physical activity as assessed by the Physical Activity Readiness Questionnaire (PAR-Q) [21]. It was anticipated that 9 groups of 20 students would be enrolled in these classes at different times across one semester (*n* = 180). Accelerometer data were collected on a sub-set of participants limited by accelerometer availability. 

For focus group interviews, purposive sampling occurred to ensure study participants represented a broad sample of the student body (various class groups, genders, campus of study and type of degree). Recruitment to the qualitative component occurred until data saturation (i.e., no new themes or data being identified with further interviews) was achieved [22]. All tutors of the subjects (*n* = 6) were also invited to participate in a focus group.

### 2.3. Intervention

Once per week, students had two, 2-h classes back-to-back on the same day (one CLASS A and one CLASS B, in variable order). Using an alternating schedule, students completed one class with movement breaks and one traditional desk-based class without movement breaks each week over a 3-week period. Classroom movement breaks were 5–10 min long [13,23] and consisted of whole-body movements (see Box 1). Tutors aimed to complete one movement break outside the classroom (walking) and two inside per class. Classroom movement breaks were incorporated after approximately 20 min of sedentary classroom time in one two-hour class per study day so that a total of 15–30 min of movement was incorporated into the two-hour class (Figure 1). Two-hour classes with no scheduled movement breaks were used as a comparison. Attempts were made to tailor the class structure to fit the breaks by modifying learning activities to be completed whilst moving, by utilizing walking discussions or by having changes in topic planned for movement break times. Movement breaks conducted in the classroom were led by the tutor or a student volunteer. Outside walking discussions were unsupervised. Participation in each movement break was voluntary and all students were invited to participate in the movement breaks regardless of study enrolment.

Box 1Movement break options.
**Example movement break activities**
Movement breaks should last between 5 and 10 min in duration. To achieve 5–10 min, a combination of these activities or games could be completed: SquatsStep LungesWall push-upsHigh knee running on the spotStar jumpsSit-to-standCalf-raisesBrisk walkingHopping or jumping Games on the website [24]

### 2.4. Outcomes

Bowen and colleagues [25] describe eight general areas of focus that may be addressed in a feasibility study: acceptability, demand, implementation, practicality, adaptation, integration, expansion and limited-efficacy testing. Bowen’s framework suggests that researchers should determine which domains to focus on depending on the purpose of the feasibility study and the study question [25]. For this study, three domains of feasibility were selected to determine if movement breaks were feasible in university classes: acceptability (levels of disruption, engagement and satisfaction), practicality (safety, ease of scheduling and conducting breaks) and limited efficacy testing of movement breaks. Engagement, ease of scheduling and conducting movement breaks and the perceived amount of disruption to the class were explored through qualitative focus group interviews with tutors and students. Safety was determined by recording adverse events such as injuries, muscle soreness, trips or falls. Limited efficacy was determined by measuring perceived concentration, alertness and enjoyment as previous research in university students has mentioned these concepts in students’ feedback on movement breaks and physically active learning [13,14,26]. Sedentary behaviour (time spent sedentary) and physical activity levels (time spent walking, time spent upright, sit-to-stand transitions and steps) were assessed to determine intervention fidelity and effectiveness. 

### 2.5. Data Collection

Focus group interviews were conducted within a week of the classes finishing for up to one hour each to determine student and tutor perceptions of movement breaks following a semi-structured interview schedule (Appendix B). Tutors were interviewed by the same facilitator (CP) separately from students so that participants felt free and comfortable talking to each other [18]. All participants were provided with information about the researcher (CP) and the project goals prior to participation. 

Perceived concentration (the ability to devote your attention to a single activity), mental alertness (a state of being ready to see and understand and being mentally aware and quick) and enjoyment (being fun and giving a sense of satisfaction) were assessed for all participants in all classes using a self-administered 10 cm visual analogue scale [27] at the end of each class (Appendix A). The visual analogue scale (VAS) has been used to evaluate a number of social, behavioural and clinical subjective phenomena because it is convenient, easily understood and quick to administer. It was chosen for this study because of the speed of administration which allowed students to move on to their next class in a timely manner. Visual analogue scales are considered valid and reliable, especially when the time between repeat measurements is small [27]. In our study with outcomes evaluated weekly, it is therefore likely that changes over time reflect actual change rather than a high degree of measurement error.

ActivPAL3 triaxial accelerometers were used to continuously monitor physical activity and sedentary behaviour during classes. Participants wore the monitors on the anterior aspect of their thigh at the central mid-thigh level to objectively assess physical activity and sedentary time to determine intervention fidelity. Students self-applied these monitors under instruction at the beginning of the first class and removed them after their second class. The activPAL3 is valid and has excellent reliability for the detection of posture and purposeful walking in young adults [28].

### 2.6. Data Analysis

Quantitative data on physical activity and sedentary behaviour were downloaded from the accelerometer at the end of each pair of classes. Intervention fidelity was assessed by comparing activity and sedentary behaviour in the control and experimental classes. Group means and standard deviations (SD) were reported and within-group mean differences (MD) and 95% confidence intervals (CI) in physical activity, sedentary behaviour and concentration, mental alertness and enjoyment between classes with and without movement breaks were assessed via paired *t*-tests using SPSS version 27 [29]. Missing data were excluded from the analysis. Two-way repeated-measures ANOVA was used to assess the interaction effects of class type and movement break condition and class time and movement break condition on the outcomes of concentration, alertness and enjoyment.

Interviews were audio-recorded and transcribed verbatim. Transcriptions were transferred to NVivo12 data management software [30]. Qualitative data were analysed thematically using an iterative process [31]. Two researchers (C.P., L.R.) independently read the transcripts in their entirety. They then re-read and commenced initial coding (open coding) independently identifying key phrases. Re-coding continued until themes emerged. The researchers then met to discuss emerging themes and looked for links between themes before deciding on the main themes. To ensure accuracy the main themes were summarised by the researchers and checked for accuracy by participants to ensure it was an accurate interpretation of their perceptions (member checking) [18]. The final step involved returning to the transcripts to selectively search for data related to the identified themes (selective coding) and to check that no perspectives had been missed. Data were collected and analysed simultaneously to guide further interviews and assess for data saturation [22].

Quantitative and qualitative data were analysed separately. The identified themes and quantitative results were then compared and contrasted in a process called triangulation for the overall synthesis [17]. As researchers bring their own beliefs and experiences into qualitative research and data interpretation, the researchers’ backgrounds are briefly described to enhance reflexivity as follows: all researchers are qualified physiotherapists and five have Doctor of Philosophy qualifications (C.P., L.H., M.D.N., A.H., G.O’D.). At the time of the intervention, five researchers were tutors in the subjects (C.P., L.H., M.D.N., A.H., J.L.) and two were students (L.R., D.M.). Three researchers are experienced in qualitative research (C.P., A.H., G.O’D.).

## 3. Results

Participants were recruited between February and March 2020. Recruitment ceased after 7 groups of students (*n* = 140) commenced the subjects due to COVID-19 restrictions on face-to-face teaching. Eighty-five students participated in the feasibility study (61% recruitment rate) and 44 wore accelerometers (52%). Accelerometer data from 43 participants were included in the analysis as one monitor malfunctioned (Figure 2). Overall, participants had similar characteristics to the total potentially eligible population, were a mean age of 23 (SD 2) years and 69% were female (Table 1).

Following purposive sampling, data saturation was reached after a total of 14 students participated in four focus group interviews. All six tutors participated in one focus group interview. The average duration of focus groups was 27 (SD 11) minutes. No corrections or additions were made after member checking.

### 3.1. Feasibility

Tutors and students perceived that engagement in movement breaks was high with almost all students participating in movement breaks (irrespective of their enrolment in the study). No adverse events were reported but tutors did remark that they actively considered safety when leading and modifying movement breaks. Tutors expressed that movement breaks did not compromise intended curriculum delivery.

#### 3.1.1. Theme 1. Movement Breaks Had Multiple Benefits

Students and tutors expressed improvements in concentration and alertness as well as enhanced rapport between students and tutors and increased camaraderie between students. These changes were perceived to then impact class participation as students described feeling more comfortable and less worried about making mistakes so that they were more willing to speak up and be interactive. More enthusiastic engagement in group tasks and higher productivity were also expressed as benefits by tutors and students.


*“I found that you know their level of engagement and their focus once they got back was higher and that certainly, they were more animated and more vocal and I think there was just generally an increased engagement afterward.”*
(Tutor 1)

Students expressed varied and differing views on when they preferred movement breaks, with some preferring movement breaks in the morning class to help rouse them whilst others felt they needed it to aid concentration during the second class after having been sedentary for the previous two hours. Depending on personal preferences related to class type (CLASS A or B), students also expressed differing views on which class they most valued the movement breaks but overall agreed it would have been best if they were in both classes.


*“I liked the mornings, I needed it then because it was the morning. Then the afternoon ones I liked it because I’d been in class for four hours [other students agreeing]”*
(Student 11)

#### 3.1.2. Theme 2. Timing Is Important to Enhance Effect and Reduce Disruption

The timing was perceived to play an important role in the effect of movement breaks. When movement breaks did not align well with a natural break in content or a change in topic, they were perceived to be somewhat disruptive:


*“In some instances, we were on a bit of a roll with the content. So then to have to stop that and do something else was counterproductive cause it took a while to get back into it. Whereas other times they timed it really well, so it was like between changeovers in topics.”*
(Student 10)

When movement breaks were timed well, they allowed students to consolidate the knowledge learned in the previous portion of the class and be reinvigorated for the next topic.


*“When the timing was good it did help to reengage the students when they were starting to get distracted and not actually making progress with the task that they were meant to be doing in class”*
(Tutor 5)

Tutors were conscious of the importance of timing and found it challenging to schedule movement breaks according to the research schedule.


*“I think I was a bit distracted from the content as well because I was so focused on time and trying to fit it in, in a way that wasn’t going to be disruptive”*
(Tutor 5)

When movement breaks were not timed well, tutors described difficulty resettling students.

#### 3.1.3. Theme 3. Some Movement Breaks Were Perceived to Be Better Than Others

The type of movement completed within the break also appeared to have an impact on perceived efficacy. Walking outdoors and competitive activities were highly regarded by students as they were perceived to give a complete mental and physical break from the classroom as opposed to, for example, balance tasks while listening to class content.


*“The ones outside of class I think [other students agreeing] were really helpful. Just staying in class and doing movement breaks, like while it was good, I felt I was like more motivated and able to hold my attention for the rest of the class if I had gone outside and came back in.”*
(Student 3)

#### 3.1.4. Theme 4. Less Frequent Movement Breaks Would Be Better

Students and tutors perceived that having less frequent movement breaks that were flexible in terms of timing to suit the class content and current energy levels of the class would have been more beneficial.


*“I found that there were too many of them. So, I thought fitting three of them in two hours was just too many. So, I felt like two or even one would have been really helpful and productive”*
(Tutor 6)


*“Probably didn’t need as many. Um, like one an hour I probably would have been happy with”*
(Student 10)

### 3.2. Limited Efficacy

#### 3.2.1. Alertness

Students reported significantly higher levels of alertness in classes with movement breaks when compared to classes without movement breaks (*p* < 0.001) (Table 2). The intervention effect remained regardless of class type or time and there was no interaction effect with either class type (*p* = 0.796) or class time (*p* = 0.194). Both students and tutors perceived that students were more alert, attentive, awake and energised after a movement break.


*“It actually genuinely does [laughter/agreement] make you more productive. You feel more alert!”*
(Student 5)

Students had higher levels of alertness in CLASS A when compared to CLASS B (*p* = 0.009), irrespective of movement breaks. Qualitative data helped explain this as tutors and students described differences in class structure:


*“I think the thing for [Class A] for me, is that when I wasn’t concentrating or alert, it’s harder to get through it. Like [Class B], you can just like, sit down and you type it down.”*
(Student 3)

#### 3.2.2. Concentration

Students reported significantly higher levels of concentration in classes with movement breaks when compared to classes without movement breaks (*p* < 0.001) (Table 2). There was no interaction effect with class type (*p* = 0.59) or time (*p* = 0.984).

Tutors perceived higher levels of concentration in their students and students expressed that movement breaks made it easier to concentrate both after a break and in the lead up to a break.


*“I get distracted very easily. I find it very hard to concentrate on any one thing for a long period of time so for me it was yeah, I think that it helped sort of wake me up and refocus me”*
(Student 11)

#### 3.2.3. Enjoyment

Students reported significantly higher levels of enjoyment in classes with movement breaks when compared to classes without movement breaks (*p* < 0.001) (Table 2). The intervention effect remained regardless of class type or time and there was no interaction effect for class (*p* = 0.61) or time (*p* = 0.232). There was a trend towards enjoyment levels being higher in the second (and final) class for the day (*p* = 0.067).

Students and tutors both perceived that classes with movement breaks were more fun and enjoyable and added laughter and humor to classes. They also allowed for social interaction and acted as icebreakers between students and tutors and amongst students.


*“Definitely made the classes more fun.”*
(Student 10)

#### 3.2.4. Sedentary Behaviour and Physical Activity

Some movement and upright time occurred during classes without movement breaks due to in-class presentations, movements for group work and students getting up from their desks to relieve discomfort. In classes with movement breaks, students spent 13 (95%CI 10 to 17) minutes less time in sedentary activities and took 834 (95%CI 675 to 994) more steps when compared to classes without movement breaks (Table 2). Students reported discomfort when sitting for prolonged periods so found that breaking up sitting was beneficial.


*“Like when I’m sitting and studying, like even just in class for like a couple of hours my body gets really sore.”*
(Student 11)

Students felt that the movement breaks most likely did not contribute to much increased physical activity overall but did report continuing to do them during their own independent study.


*“We would stay back pretty much for the rest of the day doing work. And we’d take our own movement breaks because we knew that it was helpful in class”*
(Student 6)

## 4. Discussion

This study provides preliminary evidence that it is feasible to conduct movement breaks during university classes in terms of acceptability, practicality and efficacy. Students spent less time sitting, took more steps and reported significantly higher levels of alertness, concentration, and enjoyment in classes with movement breaks when compared to classes without movement breaks. To further enhance feasibility, tutors could consider fewer movement breaks that can be more flexibly scheduled around class content and class dynamic.

Previous studies in the university classroom also found that movement breaks may be beneficial [13,14]. Blache and colleagues [14] found that compared to no break, structured exercise breaks improved vigour and reduced fatigue in 66 university students. Ferrer and Laughlin [13] found that students (*n* = 53) subjectively reported increases in attention and that movement breaks were fun. Neither study measured objective changes in physical activity. Our study adds quantitative and qualitative data to the small body of evidence that classroom movement breaks are effective in terms of cognitive and physical outcomes without compromise to curriculum delivery.

The rationale for why movement breaks improved concentration and alertness may partially be explained by the effort-recovery model [32]. The model is based on the premise that working on mentally demanding tasks (e.g., university classes) leads to fatigue, and in response to fatigue, students need to increase their effort (in stress response) to maintain their performance [33]. If this increased effort cannot be achieved or sustained, performance deteriorates [33]. The effort-recovery model [32] proposes that stress and fatigue are reversible with rest. Based on the effort-recovery model, it appears that the movement breaks conducted did allow sufficient break from the mentally demanding task to allow recovery and therefore, improvements in concentration and alertness.

The optimal dose and timing of movement breaks to produce cognitive benefits in the university classroom is still unknown. One laboratory-based randomised controlled trial [34] designed to replicate a university learning environment used 5-min exercise breaks approximately every 15–20 min. They found students who completed exercise breaks were more attentive and focused toward the end of the lecture and performed better on a quiz than students who had no break or a non-exercise break [34]. In addition, a classroom-based study reported subjective benefits to enjoyment and alertness following breaks every 20 min [13]. However, results from our feasibility study indicate students and tutors felt movement breaks every 20 min were too frequent. Another classroom-based study conducted a movement break after 45 min of sedentary time and reported benefits remained present upon testing 20 min later [14]. These results are promising and suggest that less frequent breaks may still be effective due to a lasting influence while also being more practical. However, Niedermeier and colleagues [35] evaluated the effect of a 10-min outdoor running break 45 min into a 2-h study course. They found the running break improved visual attention, perceived attention and alertness immediately following the break compared to the control group but there were no differences in enjoyment, and effects on attention and alertness were not maintained 30 min later. Perhaps the less intense movement breaks conducted in our feasibility study allowed more interaction and increased enjoyment and therefore may have longer-lasting effects. Considering the results of our feasibility study and previous studies with university students [13,14,34,35], future research may like to focus on the effects of breaks after 30–45 min sedentary time and monitor the duration of the benefit.

Students in our study and a previous study [13] expressed that movement breaks were enjoyable. As student satisfaction is increasingly important to universities because of its impact on student recruitment and retention [36], anything that can help with this is a positive. In addition, enjoyment may be related to academic outcomes as Pekrun’s three-dimension emotional taxonomy categorises enjoyment of an activity as a positive activating emotion which is associated with better student outcomes [37]. An unintended consequence of movement breaks being enjoyable was that they enhanced comradery between students and increased rapport between students and tutors. As a result, students expressed feeling more able to speak up and engage in classes. Through enjoyment and engagement, movement breaks may further enhance learning and university outcomes.

The levels of physical activity completed in movement breaks are only a fraction of what is recommended in daily physical activity guidelines, however, if movement break practices flow into students’ own personal study the impact may be magnified. The negative effect of high-volume sedentary behaviour (such as that seen on weekdays in university students) may be independent of physical inactivity levels [38], however, those who participate in moderate-vigorous intensity exercise may mitigate some risk [39]. Students in this study did report that they changed their study habits following participation in class which may increase the potential for reducing risks related to high volume sedentary time. Future research could involve advice and education on breaking up sedentary time outside the classroom and assess carry-over to determine the overall impact on physical activity and sedentary behaviour.

The results from this study, combined with results from other studies in university students [13,14,34,35] suggest that movement breaks may be beneficial to university students in terms of enhancing concentration, alertness and enjoyment, reducing sedentary behaviour and increasing physical activity. University educators should consider incorporating movement breaks after 30–45 min of sedentary time in classes of more than 1-h duration. The benefits of these movement breaks have the potential to translate into improved academic performance, enhanced student satisfaction and may improve the health and wellbeing of university students. Possible barriers to implementation that need to be considered include having adequate space, considering safety, structuring breaks around class content so they do not disrupt the flow and having tutors/lecturers who are comfortable leading movement breaks. To facilitate implementation, university educators can refer to existing movement break options [24], consider natural breaks in the classes where there is a change in topic or focus and consider opportunities for walking small group discussions when students are asked to brainstorm or discuss a topic.

This study included both qualitative and quantitative data and included the views of students and tutors to determine if it was feasible to conduct movement breaks during university classes. Although most students participated in the classroom movement breaks, results are only generalisable to physiotherapy students who volunteered to participate and complete the outcomes which may have introduced participation bias. In addition, the study needed to be stopped early due to COVID-19 restrictions which limited recruitment. The reliability and validity of the VAS in relation to alertness, concentration and enjoyment have not been established in university students and, therefore, these results should be interpreted with caution. However, the performance properties of VAS scales, in general, are well established and our results were triangulated with qualitative interviews to increase the trustworthiness and consistency of the findings. Although not a randomised controlled trial, this feasibility study did include control classes for comparison. Another strength of the study was that it was conducted pragmatically during usual university classes so that results are relevant to those wanting to implement movement breaks in their classes.

## 5. Conclusions

This mixed-methods study provides preliminary evidence that classroom movement breaks are feasible, reduce sedentary behaviour and increase physical activity, alertness, concentration and enjoyment in university students. Classroom movement breaks may further improve academic performance through enhanced engagement and may further impact physical activity and sedentary behaviour through independent carry-over of habits. Results suggest that outdoor or competitive movement breaks were the most well-received by students and that a frequency of less than every 20 min may be more practical.

## Figures and Tables

**Figure 1 ijerph-18-05589-f001:**
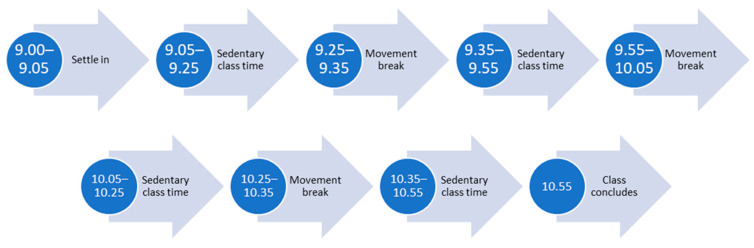
Example class schedule.

**Figure 2 ijerph-18-05589-f002:**
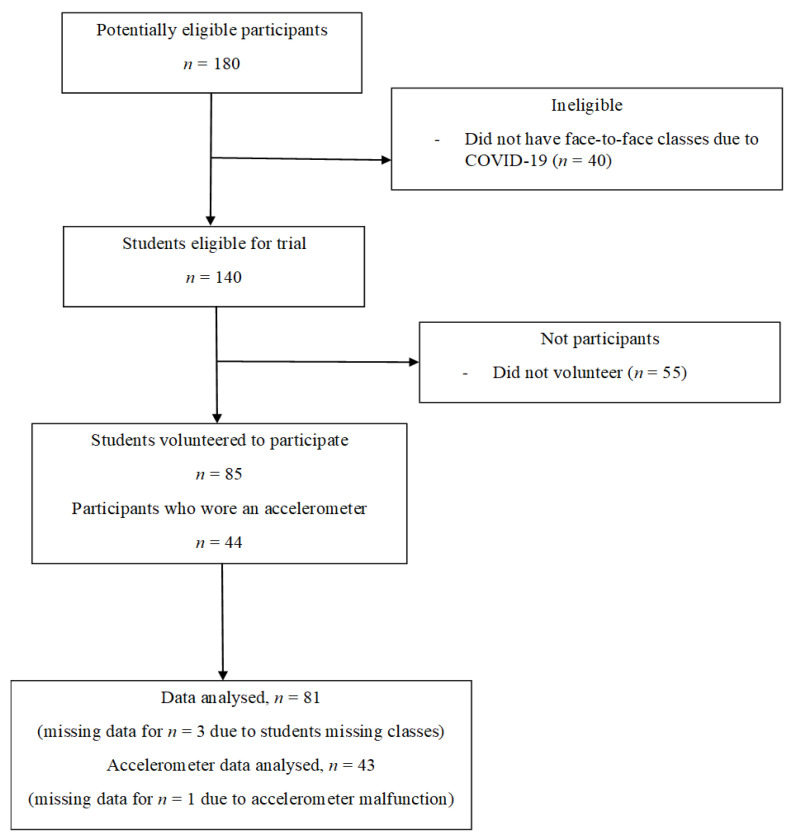
Flow of participants through the study.

**Table 1 ijerph-18-05589-t001:** Participant characteristics.

Participant Characteristics	All Student Participants	Focus Group—Students	Focus Group—Tutors	Accelerometer Wearers
*n* =	85	14	6	44
Gender				
Male	26	2	3	13
Female	58	12	3	30
Non-binary/Gender fluid/ Queer	1	0	0	1
Enrolment				
Bachelor/Master degree	66	10	NA	34
Graduate Entry Masters	19	4	NA	10
Domestic student	77	14	NA	39
International student	8	0	NA	5
Location				
Metropolitan campus	62	12	4	44
Regional campus	23	2	2	0

**Table 2 ijerph-18-05589-t002:** Limited efficacy testing outcomes.

Class	Movement Break, Mean (SD)	No Movement Break, Mean (SD)	Difference, Mean, 95% CI	*p*-Value
Alertness (*n* = 81)				
CLASS A	7.0 (1.5)	5.5 (1.7)	1.5 (1.1 to 1.9)	<0.001
CLASS B	6.6 (1.5)	5.2 (1.8)	1.4 (1.0 to 1.8)	<0.001
Concentration (*n* = 81)				
CLASS A	6.8 (1.6)	5.4 (1.8)	1.4 (1.0 to 1.9)	<0.001
CLASS B	6.7 (1.6)	5.0 (1.8)	1.6 (1.2 to 2.1)	<0.001
Enjoyment (*n* = 81)				
CLASS A	7.3 (1.6)	5.6 (1.8)	1.7 (1.3 to 2.2)	<0.001
CLASS B	7.0 (2.0)	5.5 (1.7)	1.6 (1.1 to 2.0)	<0.001
Sedentary behaviour and physical activity (*n* = 43)				
Seated time (minutes)	96 (10)	109 (10)	−13 (−17 to −10)	<0.001
Steps taken	992 (477)	157 (177)	834 (675 to 994)	<0.001
Sit-to-stand transitions	6 (2)	3 (3)	2 (1–3)	<0.001
Standing time (minutes)	11 (6)	8 (9)	3 (0 to 6)	0.024
Walking time (minutes)	12 (6)	2 (2)	10 (8 to 12)	<0.001

## Data Availability

The data presented in this study is available on request from the corresponding author. The data are not publicly available due to ethics limitations.

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
