# Peer review of "Classroom Movement Breaks Reduce Sedentary Behavior and Increase Concentration, Alertness and Enjoyment during University Classes: A Mixed-Methods Feasibility Study"

_ijerph, 2021, doi:10.3390/ijerph18115589_

Round 1

Reviewer 1 Report

This paper focuses on an important but often ignored measure to improve both the student’s well-being and class performance. This mixed study provides sufficient evidence that it is feasible to conduct movement breaks. This paper provided sufficient information about the study, the methods used are sound, and the conclusions are reliable.

Figure 2 needs some editing.

Author Response

Response: Thank you. The Figure 2 formatting issues have now been corrected.

Reviewer 2 Report

Dear authors,

I enjoyed reading your well-written manuscript. It is an intersting topic and I only have some comments.

Would you mind to include the quantitative questionnaire as online supplement?

I know that this study has several strenghts. Nonetheless, some weaknesses should be discussed. These include a potential participation bias due to those who did not like to participate and those who could not due to Covid-19. Although generalizability may be limited.

I think it would be interesting to further point out the benefits of breaks with sport inside the classroom. For instance, Niedermeier (2020; IJERPH 17, 3678) conduct studies on breaks outside the classroom (I have reviewed them once). A clear strengths are breaks with physical activity that can be conducted within the classroom. Niedermeier and colleages let sport students run outside for 10 minutes. I do not know how feasible this is. So maybe, you are able to describe in more detail differences between breaks inside and outside the classroom. This would be helpful also for universities and tutors that aim to include breaks in their curriculum.

Author Response

Thank you for your comments and suggestions. Below you will find an itemized response to each comment outlining the changes we have made which we feel have strengthened and improved out manuscript.

1) I enjoyed reading your well-written manuscript. It is an interesting topic and I only have some comments. Would you mind to include the quantitative questionnaire as online supplement?

Response: As suggested, we have provided the participant questionnaire as supplemental file 1.

2) I know that this study has several strengths. Nonetheless, some weaknesses should be discussed. These include a potential participation bias due to those who did not like to participate and those who could not due to Covid-19. Although generalizability may be limited.

Response: as suggested, we have added discussion of these to the limitations section as follows:

“Although most students participated in the classroom movement breaks, results are only generalizable to physiotherapy students who volunteered to participate and complete the outcomes which may be have introduced participation bias. In addition, the study needed to be stopped early due to COVID-19 restrictions which limited recruitment.”

3) I think it would be interesting to further point out the benefits of breaks with sport inside the classroom. For instance, Niedermeier (2020; IJERPH 17, 3678) conduct studies on breaks outside the classroom (I have reviewed them once). A clear strengths are breaks with physical activity that can be conducted within the classroom. Niedermeier and colleages let sport students run outside for 10 minutes. I do not know how feasible this is. So maybe, you are able to describe in more detail differences between breaks inside and outside the classroom. This would be helpful also for universities and tutors that aim to include breaks in their curriculum.

Response: Thank you for pointing out this interesting and relevant paper. We now refer to it to enhance our discussion of movement break timing as follows:

“However, Niedermeier and colleagues [35] evaluated the effect of a 10-minute outdoor running break 45 minutes into a 2-hour study course. They found the running break improved visual attention, perceived attention and alertness immediately following the break compared to the control group but there were no differences in enjoyment and effects on attention and alertness were not maintained 30 minutes later. Perhaps the less intense movement breaks conducted in our feasibility study allowed more interaction and increased enjoyment and therefore may have longer lasting effects.”

“Considering the results of our study and previous studies [13, 14, 34, 35], future research may like to focus on the effects of breaks after 30-45 minutes sedentary time and monitor the duration of the benefit.”

Reviewer 3 Report

Congratulations on the study, it seems to me a very interesting and complete study. I recommend a series of improvements for its publication:
Provide what implications this study has for university practice.
Describe the variables as they have been collected. Perhaps expand the section on Outcomes. How have these variables been measured?
Improve Figure 2.

Author Response

Thank you for your comments and suggestions. Below you will find an itemized response to each comment outlining the changes we have made which we feel have strengthened and improved our manuscript.

1) Congratulations on the study, it seems to me a very interesting and complete study. I recommend a series of improvements for its publication:
Provide what implications this study has for university practice.

Response: As suggested, we have added a paragraph on implications for universities:

“The results from this study, combined with results from other studies in university students [13, 14, 34, 35] suggest that movement breaks may be beneficial to university students in terms of enhancing concentration, alertness and enjoyment, reducing sedentary behavior and increasing physical activity. University educators should consider incorporating movement breaks after 30-45 minutes of sedentary time in classes of more than 1-hour duration. The benefits of these movement breaks have the potential to translate into improved academic performance, enhanced student satisfaction and may improve the health and wellbeing of university students. Possible barriers to implementation that need to be considered include having adequate space, considering safety, structuring breaks around class content so they don’t disrupt the flow and having tutors/lecturers who are comfortable to lead movement breaks. To facilitate implementation, university educators can refer to existing movement break options [24], consider natural breaks in the classes where there is a change in topic or focus and consider opportunities for walking small group discussions when students are asked to brain-storm or discuss a topic”.      

2) Describe the variables as they have been collected. Perhaps expand the section on Outcomes. How have these variables been measured?

Response: As suggested we have expanded the outcomes and data collection sections to improve clarity.

Bowen’s framework was described in more detail: “Bowen and colleagues [25] describe eight general areas of focus that may be addressed in a feasibility study: acceptability, demand, implementation, practicality, adaptation, integration, expansion and limited-efficacy testing. Bowen’s framework suggests that researchers should determine which domains to focus on depending on the purpose of the feasibility study and the study question [25]. For this study, three domains of feasibility were selected to determine if movement breaks were feasible in university classes: acceptability (levels of disruption, engagement and satisfaction), practicality (safety, ease of scheduling and conducting breaks) and limited efficacy testing.”

Outcome measures are also described in more detail: “The visual analogue scale (VAS) has been used to evaluate a number of social, behavioural and clinical subjective phenomena because it is convenient, easily understood and quick to administer. It was chosen for this study because of the speed of administration which allowed students to move on to their next class in a timely manner. Visual analogue scales are considered valid and reliable, especially when the time between repeat measurements is small [27]. In our study with outcomes evaluated weekly it is therefore likely that changes over time reflect actual change rather than a high degree of measurement error”.

3) Improve Figure 2.

Response: The Figure 2 formatting issues have now been corrected.

Reviewer 4 Report

I appreciate the authors for addressing the very interesting issue of the feasibility of incorporating movement breaks into university classes in terms of acceptability , practicality  and efficacy. I appreciate the  authors' research approach, the use of qualitative and quantitative methods and that it is a feasibility study. I agree with  the authors that this type of study (hich evidence that classroom movement breaks are effective in terms of cognitive and physical outcomes) is lacking in the literature. We  have many research findings on the effect of physical activity on  students' well-being, or the effect of study breaks on students' general well-being. At the same time, I  see several weaknesses in the manuscript  that need to be supplemented/clarified:

1) To what extent can the survey results be generalised to the student  population? To what extent can the results obtained in a selected  academic environment be considered representative? Furthermore, I think  it is a good idea to introduce the context of the survey in the paper.  It will be more understandable to the international reader who will be generally introduced to the educational system issues in this case.

2)  The authors indicate possible future research directions but do not  describe the limitations of this study. These limitations are related, among other things, to  the research tools used. The use of the VAS in the behavioural sciences  is common, e.g. in medicine and psychology. It has its advantages, but at the same time some limitations, e.g. the disadvantage of the VAS is  that it is one-dimensional. To what extent can such measurements be  considered reliable? In addition, the authors should describe in more  detail the variables that were analysed, together with a description of  the tool used to measure them.

3) Could classroom movement breaks be a common practice? I expect the authors to raise in the discussion the question of possible barriers that may arise in the application of research findings to academic practice.

Overall I think the authors have conducted valuable research that may  be of interest to the journal reader.

Author Response

Thank you for your comments and suggestions. Below you will find an itemized response to each comment outlining the changes we have made which we feel have strengthened and improved our manuscript.

1) To what extent can the survey results be generalised to the student  population? To what extent can the results obtained in a selected  academic environment be considered representative?

Response: To address this comment we have added some discussion of generalizability to the limitations section of the discussion as follows: “Although most students participated in the classroom movement breaks, results are only generalizable to physiotherapy students who volunteered to participate and complete the outcomes which may have introduced participation bias. In addition, the study needed to be stopped early due to COVID-19 restrictions which limited recruitment.”

2) Furthermore, I think  it is a good idea to introduce the context of the survey in the paper.  It will be more understandable to the international reader who will be generally introduced to the educational system issues in this case.

Response: As suggested we have now provided the participant questionnaire as supplemental file 1 to improve clarity.

3) The authors indicate possible future research directions but do not  describe the limitations of this study. These limitations are related, among other things, to  the research tools used. The use of the VAS in the behavioural sciences  is common, e.g. in medicine and psychology. It has its advantages, but at the same time some limitations, e.g. the disadvantage of the VAS is  that it is one-dimensional. To what extent can such measurements be  considered reliable?

Response: As suggested, we have added generalizability, participation bias (as above) and outcomes used to the limitations section of the discussion: “The reliability and validity of the VAS in relation to alertness, concentration and enjoyment has not been established in university students and therefore these results should be interpreted with caution. However, the performance properties of VAS scales in general are well established and our results were triangulated with qualitative interviews to increase trustworthiness and consistency of the findings”.

We have also discussed the reliability and validity of the VAS in the data collection section: “The visual analogue scale (VAS) has been used to evaluate a number of social, behavioural and clinical subjective phenomena because it is convenient, easily understood and quick to administer. It was chosen for this study because of the speed of administration which allowed students to move on to their next class in a timely manner. Visual analogue scales are considered valid and reliable, especially when the time between repeat measurements is small [27]. In our study with outcomes evaluated weekly it is therefore likely that changes over time reflect actual change rather than a high degree of measurement error”.

3) In addition, the authors should describe in more  detail the variables that were analysed, together with a description of  the tool used to measure them.

Response: To improve clarity around outcomes assessed, we have expanded the outcomes section of the methods by adding the data collection survey as supplementary material and providing more information on the VAS (as above).

4) Could classroom movement breaks be a common practice? I expect the authors to raise in the discussion the question of possible barriers that may arise in the application of research findings to academic practice.

Response: Thank you. We have now added a paragraph to the discussion on implications of the findings for universities which includes possible barriers: “The results from this study, combined with results from other studies in university students [13, 14, 34, 35] suggest that movement breaks may be beneficial to university students in terms of enhancing concentration, alertness and enjoyment, reducing sedentary behavior and increasing physical activity. University educators should consider incorporating movement breaks after 30-45 minutes of sedentary time in classes of more than 1-hour duration. The benefits of these movement breaks have the potential to translate into improved academic performance, enhanced student satisfaction and may improve the health and wellbeing of university students. Possible barriers to implementation that need to be considered include having adequate space, considering safety, structuring breaks around class content so they don’t disrupt the flow and having tutors/lecturers who are comfortable to lead movement breaks. To facilitate implementation, university educators can refer to existing movement break options [24], consider natural breaks in the classes where there is a change in topic or focus and consider opportunities for walking small group discussions when students are asked to brain-storm or discuss a topic”.      

Round 2

Reviewer 4 Report

The authors have satisfactorily responded to all my questions and made the necessary changes to the manuscript.